# Sample Efficient Imitation Learning for Continuous Control

**Fumihiro Sasaki, Tetsuya Yohira & Atsuo Kawaguchi**
Ricoh Company, Ltd.
{fumihiro.fs.sasaki,tetsuya.yohira,atsuo.kawaguchi}@jp.ricoh.com

## Abstract

The goal of imitation learning (IL) is to enable a learner to imitate expert behavior given expert demonstrations. Recently, generative adversarial imitation learning (GAIL) has shown significant progress on IL for complex continuous tasks. However, GAIL and its extensions require a large number of environment interactions during training. In real-world environments, the more an IL method requires the learner to interact with the environment for better imitation, the more training time it requires, and the more damage it causes to the environments and the learner itself. We believe that IL algorithms could be more applicable to real-world problems if the number of interactions could be reduced. In this paper, we propose a model-free IL algorithm for continuous control. Our algorithm is made up mainly three changes to the existing adversarial imitation learning (AIL) methods – (a) adopting off-policy actor-critic (Off-PAC) algorithm to optimize the learner policy, (b) estimating the state-action value using off-policy samples without learning reward functions, and (c) representing the stochastic policy function so that its outputs are bounded. Experimental results show that our algorithm achieves competitive results with GAIL while significantly reducing the environment interactions.

## 1 Introduction

Recent advances in reinforcement learning (RL) have achieved super-human performance on several domains (Mnih et al., 2015; Silver et al., 2016; Mnih et al., 2016; Lillicrap et al., 2015). On most of such domains with the success of RL, the design of reward, that explains what agent's behavior is favorable, is obvious for humans. Conversely, on domains where it is unclear how to design the reward, agents trained by RL algorithms often obtain poor policies and behave worse than what we expect them to do. Imitation learning (IL) comes in such cases. The goal of IL is to enable the learner to imitate expert behavior given the expert demonstrations without the reward signal. We are interested in IL because we desire an algorithm that can be applied to real-world problems for which it is often hard to design the reward. In addition, since it is generally hard to model a variety of real-world environments with an algorithm, and the state-action pairs in a vast majority of real-world applications such as robotics control can be naturally represented in continuous spaces, we focus on model-free IL for continuous control.

A wide variety of IL methods have been proposed in the last few decades. The simplest IL method among those is behavioral cloning (BC) (Pomerleau, 1991) which learns an expert policy in a supervised fashion without environment interactions during training. BC can be the first IL option when enough demonstration is available. However, when only a limited number of demonstrations are available, BC often fails to imitate the expert behavior because of the problem which is referred to *compounding error* (Ross & Bagnell, 2010) – inaccuracies compound over time and can lead the learner to encounter unseen states in the expert demonstrations. Since it is often hard to obtain a large number of demonstrations in real-world environments, BC is often not the best choice for real-world IL scenarios.

Another widely used approach, which overcomes the compounding error problem, is Inverse Reinforcement Learning (IRL) (Russell, 1998; Ng & Russell, 2000; Abbeel & Ng, 2004; Ziebart et al., 2008). Recently, Ho & Ermon (2016) have proposed generative adversarial imitation learning

(GAIL) which is based on prior IRL works. Since GAIL has achieved state-of-the-art performance on a variety of continuous control tasks, the adversarial IL (AIL) framework has become a popular choice for IL (Baram et al., 2017; Hausman et al., 2017; Li et al., 2017). It is known that the AIL methods are more sample efficient than BC in terms of the expert demonstration. However, as pointed out by Ho & Ermon (2016), the existing AIL methods have sample complexity in terms of the environment interaction. That is, even if enough demonstration is given by the expert before training the learner, the AIL methods require a large number of state-action pairs obtained through the interaction between the learner and the environment[1]. The sample complexity keeps existing AIL from being employed to real-world applications for two reasons. First, the more an AIL method requires the interactions, the more training time it requires. Second, even if the expert safely demonstrated, the learner may have policies that damage the environments and the learner itself during training. Hence, the more it performs the interactions, the more it raises the possibility of getting damaged. For the real-world applications, we desire algorithms that can reduce the number of interactions while keeping the imitation capability satisfied as well as the existing AIL methods do.

The following three properties of the existing AIL methods which may cause the sample complexity in terms of the environment interactions:

(a) Adopting on-policy RL methods which fundamentally have sample complexity in terms of the environment interactions.

(b) Alternating three optimization processes — learning reward functions, value estimation with learned reward functions, and RL to update the learner policy using the estimated value. In general, as the number of parameterized functions which are related to each other increases, the training progress may be unstable or slower, and thus more interactions may be performed during training.

(c) Adopting Gaussian policy as the learner's stochastic policy, which has infinite support on a continuous action space. In common IL settings, we observe action space of the expert policy from the demonstration where the expert action can take on values within a bounded (finite) interval. As Chou & Scherer. (2017) suggests, the policy which can select actions outside the bound may slow down the training progress and make the problem harder to solve, and thus more interactions may be performed during training.

In this paper, we propose an IL algorithm for continuous control to improve the sample complexity of the existing AIL methods. Our algorithm is made up mainly three changes to the existing AIL methods as follows:

(a) Adopting off-policy actor-critic (Off-PAC) algorithm (Degris et al., 2012) to optimize the learner policy instead of on-policy RL algorithms. Off-policy learning is commonly known as the promising approach to improve the complexity.

(b) Estimating the state-action value using off-policy samples without learning reward functions instead of using on-policy samples with the learned reward functions. Omitting the reward learning reduces functions to be optimized. It is expected to make training progress stable and faster and thus reduce the number of interactions during training.

(c) Representing the stochastic policy function of which outputs are bounded instead of adopting Gaussian policy. Bounding action values may make the problem easier to solve and make the training faster, and thus reduce the number of interactions during training.

Experimental results show that our algorithm enables the learner to imitate the expert behavior as well as GAIL does while significantly reducing the environment interactions. Ablation experimental results show that (a) adopting the off-policy scheme requires about 100 times fewer environment interactions to imitate the expert behavior than the one on-policy IL algorithms require, (b) omitting the reward learning makes the training stable and faster, and (c) bounding action values makes the training faster.

---

[1]Throughout this paper, we refer to "number of interactions" as the number of state-action pairs obtained through interaction between the learner and the environment during training the learner.

## 2 BACKGROUND

### 2.1 PRELIMINARIES

We consider a Markov Decision Process (MDP) which is defined as a tuple $\{\mathcal{S}, \mathcal{A}, \mathcal{T}, R, d_0, \gamma\}$, where $\mathcal{S}$ is a set of states, $\mathcal{A}$ is a set of possible actions agents can take, $\mathcal{T} : \mathcal{S} \times \mathcal{A} \times \mathcal{S} \to [0, 1]$ is a transition probability, $R : \mathcal{S} \times \mathcal{A} \to \mathbb{R}$ is a reward function, $d_0 : \mathcal{S} \to [0, 1]$ is a distribution over initial states, and $\gamma \in [0, 1)$ is a discount factor. The agent's behavior is defined by a stochastic policy $\pi : \mathcal{S} \times \mathcal{A} \to [0, 1]$ and $\Pi$ denotes a set of the stochastic policies. We denote $\mathcal{S}_E \subset \mathcal{S}$ and $\mathcal{A}_E \subset \mathcal{A}$ as sets of states and actions observed in the expert demonstration, and $\mathcal{S}_\pi \subset \mathcal{S}$ and $\mathcal{A}_\pi \subset \mathcal{A}$ as sets of those observed in rollouts following a policy $\pi$. We will use $\pi_E, \pi_\theta, \beta \in \Pi$ to refer to the expert policy, the learner policy parameterized by $\theta$, and a behavior policy, respectively. Given a policy $\pi$, performance measure of $\pi$ is defined as $\mathcal{J}(\pi, R) = \mathbb{E}\big[\sum_{t=0}^{\infty} \gamma^t R(s_t, a_t) | d_0, \mathcal{T}, \pi\big]$ where $s_t \in \mathcal{S}$ is a state that the agent receives at discrete time-step $t$, and $a_t \in \mathcal{A}$ is an action taken by the agent after receiving $s_t$. The performance measure indicates expectation of the discounted return $\sum_{t=0}^{\infty} \gamma^t R(s_t, a_t)$ when the agent follows the policy $\pi$ in the MDP. Using discounted state visitation distribution denoted by $\rho_\pi(s) = \sum_{t=0}^{\infty} \gamma^t \mathbb{P}(s_t = s | d_0, \mathcal{T}, \pi)$ where $\mathbb{P}$ is a probability that the agent receives the state $s$ at time-step $t$, the performance measure can be rewritten as $\mathcal{J}(\pi, R) = \mathbb{E}_{s \sim \rho_\pi, a \sim \pi}\big[R(s, a)\big]$. The state-action value function for the agent following $\pi$ is defined as $Q_\pi(s_t, a_t) = \mathbb{E}\big[\sum_{u=t}^{\infty} \gamma^{u-t} R(s_u, a_u) | \mathcal{T}, \pi\big]$, and $Q_{\pi, \nu}$ denotes its approximator parameterized by $\nu$.

### 2.2 ADVERSARIAL IMITATION LEARNING

We briefly describe objectives of RL, IRL, and AIL below. We refer the readers to Ho & Ermon (2016) for details. The goal of RL is to find an optimal policy that maximizes the performance measure. Given the reward function $R$, the objective of RL with parameterized stochastic policies $\pi_\theta : \mathcal{S} \times \mathcal{A} \to [0, 1]$ is defined as follows:

$$\text{RL}(R) = \arg\max_\theta \mathcal{J}(\pi_\theta, R) \tag{1}$$

The goal of IRL is to find a reward function based on an assumption that the discounted returns earned by the expert behavior are greater than or equal to those earned by any non-experts behavior. Technically, the objective of IRL is to find reward functions $R_\omega : \mathcal{S} \times \mathcal{A} \to \mathbb{R}$ parameterized by $\omega$ that satisfies $\mathcal{J}(\pi_E, R_\omega) \geq \mathcal{J}(\pi, R_\omega)$ where $\pi$ denotes the non-expert policy. The existing AIL methods adopt max-margin IRL (Abbeel & Ng, 2004) of which objective can be defined as follows:

$$\text{IRL}(\pi_E) = \arg\max_\omega \mathcal{J}(\pi_E, R_\omega) - \mathcal{J}(\pi, R_\omega) \tag{2}$$

The objective of AIL can be defined as a composition of the objectives (1) and (2) as follows:

$$\text{AIL}(\pi_E) = \arg\min_\theta \arg\max_\omega \mathcal{J}(\pi_E, R_\omega) - \mathcal{J}(\pi_\theta, R_\omega) \tag{3}$$

### 2.3 OFF-POLICY ACTOR-CRITIC

The objective of Off-PAC to train the learner can be described as follows:

$$\arg\max_\theta \mathbb{E}_{s \sim \rho_\beta, a \sim \pi_\theta}\big[Q_{\pi_\theta, \nu}(s, a)\big] \tag{4}$$

The learner policy is updated by taking the gradient of the state-action value. Degris et al. (2012) proposed the gradient as follows:

$$\mathbb{E}_{s \sim \rho_\beta, a \sim \pi_\theta}\big[Q_{\pi_\theta, \nu}(s, a) \nabla_\theta \log \pi_\theta(a|s)\big] \tag{5}$$

Heess et al. (2015) provided another formula of the gradient using "re-parameterization trick" in the case that the learner policy selects the action as $a = \pi_\theta(s, z)$ with random variables $z \sim P_z$ generated by a distribution $P_z$:

$$\mathbb{E}_{s \sim \rho_\beta, z \sim P_z}\big[\nabla_a Q_{\pi_\theta, \nu}(s, a)|_{a = \pi_\theta(s, z)} \nabla_\theta \pi_\theta(s, z)\big] \tag{6}$$

## 3 ALGORITHM

As mentioned in Section.1, our algorithm (a) adopts Off-PAC algorithms to train the learner policy, (b) estimates state-action value without learning the reward functions, and (c) represents the stochastic policy function so that its outputs are bounded. In this section, we first introduce (b) in 3.1 and describe (c) in 3.2, then present how to incorporate (b) and (c) into (a) in 3.3.

## 3.1 VALUE ESTIMATION WITHOUT REWARD LEARNING

In this subsection, we introduce a new IRL objective to learn the reward function in 3.1.1 and a new objective to learn the value function approximator in 3.1.2. Then, we show that combining those objectives derives a novel objective to learn the value function approximator without reward learning in 3.1.3.

### 3.1.1 REWARD LEARNING

We define the parameterized reward function as $R_\omega(s, a) = \log r_\omega(s, a)$, with a function $r_\omega : \mathcal{S} \times \mathcal{A} \to [0, 1]$ parameterized by $\omega$. $r_\omega(s, a)$ represents a probability that the state-action pairs $(s, a)$ belong to $\mathcal{S}_E \times \mathcal{A}_E$. In other words, $r_\omega(s, a)$ explains how likely the expert executes the action $a$ at the state $s$. With this reward, we can also define a Bernoulli distribution $p_\omega : \Pi \times \mathcal{S} \times \mathcal{A} \to [0, 1]$ such that $p_\omega(\pi_E|s, a) = r_\omega(s, a)$ for the expert policy $\pi_E$ and $p_\omega(\pi|s, a) = 1 - r_\omega(s, a)$ for any other policies $\pi \in \Pi \setminus \{\pi_E\}$ which include $\pi_\theta$ and $\beta$. A nice property of this definition of the reward is that the discounted return for a trajectory $\{(s_0, a_0), (s_1, a_1), ...\}$ can be written as a log likelihood with $p_\omega(\pi_E|s_t, a_t)$:

$$\sum_{t=0}^{\infty} \gamma^t R_\omega(s_t, a_t) = \log \prod_{t=0}^{\infty} r_\omega^{\gamma^t}(s_t, a_t) = \log \prod_{t=0}^{\infty} p_\omega^{\gamma^t}(\pi_E|s_t, a_t) \tag{7}$$

Here, we assume Markov property in terms of $p_\omega$ such that $p_\omega(\pi_E|s_t, a_t)$ for $t \geq 1$ is independent of $p_\omega(\pi_E|s_{t-u}, a_{t-u})$ for $u \in \{1, ..., t\}$. Under this assumption, the return naturally represents how likely a trajectory is the one the expert demonstrated. The discount factor $\gamma$ plays a role to make sure the return is finite as in standard RL.

The IRL objective (2) can be said to aim at assigning $r_\omega = 1$ for state-action pairs $(s, a) \in \mathcal{S}_E \times \mathcal{A}_E$ and $r_\omega = 0$ for $(s, a) \in \mathcal{S}_\pi \times \mathcal{A}_\pi$ when the same definition of the reward $R_\omega(s, a) = \log r_\omega(s, a)$ is used. Following this fashion easily leads to a problem where the return earned by the non-expert policy becomes $-\infty$, since $\log r_\omega(s, a) = -\infty$ if $r_\omega(s, a) = 0$ and thus $\log \prod_{t=0}^{\infty} r_\omega^{\gamma^t}(s_t, a_t) = -\infty$ for $(s, a) \in \mathcal{S}_\pi \times \mathcal{A}_\pi$. The existing AIL methods seem to mitigate this problem by trust region optimization for parameterized value function approximator (Schulman et al., 2015b), and it works somehow. However, we think this problem should be got rid of in a fundamental way. We propose a different approach to evaluate state-action pairs $(s, a) \in \mathcal{S}_\pi \times \mathcal{A}_\pi$. Intuitively, the learner does not know how the expert behaves in the states $s \in \mathcal{S} \setminus \mathcal{S}_E$ — that is, it is uncertain which actions the expert executes in the states the expert has not visited. We thereby define a new IRL objective as follows:

$$\arg\max_\omega \mathbb{E}_{s \sim \rho_{\pi_E}, a \sim \pi_E}[p_\omega(\pi_E|s, a)] + \mathbb{E}_{s \sim \rho_\pi, a \sim \pi}[H(p_\omega(\cdot|s, a))] \tag{8}$$

where $H$ denotes entropy of Bernoulli distribution such that:

$$H(p_\omega(\cdot|s, a)) = -p_\omega(\pi_E|s, a)\log p_\omega(\pi_E|s, a) - p_\omega(\pi|s, a)\log p_\omega(\pi|s, a) \tag{9}$$

Unlike the existing AIL methods, our IRL objective is to assign $p_\omega(\pi_E|s, a) = p_\omega(\pi|s, a) = 0.5$ for $(s, a) \in \mathcal{S}_\pi \times \mathcal{A}_\pi$. This uncertainty $p_\omega(\pi_E|s, a) = 0.5$ explicitly makes the return earned by the non-expert policy finite. On the other hand, the objective is to assign $r_\omega = 1$ for $(s, a) \in \mathcal{S}_E \times \mathcal{A}_E$ as do the existing AIL methods. The optimal solution for the objective (8) satisfies the assumption of IRL : $\mathcal{J}(\pi_E, R_\omega) \geq \mathcal{J}(\pi, R_\omega)$, even though the objective does not aim at discriminating between $(s, a) \in \mathcal{S}_E \times \mathcal{A}_E$ and $(s, a) \in \mathcal{S}_\pi \times \mathcal{A}_\pi$,

### 3.1.2 VALUE FUNCTION ESTIMATION

As we see in Equation (7), the discounted return can be represented as a log likelihood. Therefore, a value function approximator $Q_{\pi_\theta}$ following the learner policy $\pi_\theta$ can be formed as a log probability. We introduce a function $q_{\pi_\theta,\nu} : \mathcal{S} \times \mathcal{A} \to [0, 1]$ parameterized by $\nu$ to represent the approximator $Q_{\pi_\theta,\nu}$ as follows:

$$Q_{\pi_\theta,\nu}(s_t, a_t) = \log q_{\pi_\theta,\nu}(s_t, a_t) \tag{10}$$

The optimal value function following a policy $\pi$ satisfies the Bellman equation $Q_\pi(s_t, a_t) = R(s_t, a_t) + \gamma \mathbb{E}_{s_{t+1} \sim \mathcal{T}, a_{t+1} \sim \pi}[Q_\pi(s_{t+1}, a_{t+1})]$. Substituting $\pi_\theta$ for $\pi$, $\log r_\omega(s_t, a_t)$ for $R(s_t, a_t)$,

and $\log q_{\pi_\theta,\nu}(s_t, a_t)$ for $Q_\pi(s_t, a_t)$, the Bellman equation for the learner policy $\pi_\theta$ can be written as follows:

$$\log q_{\pi_\theta,\nu}(s_t, a_t) = \mathbb{E}_{s_{t+1}\sim\mathcal{T}, a_{t+1}\sim\pi_\theta}\left[\log r_\omega(s_t, a_t) q_{\pi_\theta,\nu}^\gamma(s_{t+1}, a_{t+1})\right] \tag{11}$$

We introduce additional Bernoulli distributions $P_\nu : \Pi \times \mathcal{S} \times \mathcal{A} :\to [0, 1]$ and $P_{\omega\nu\gamma} : \Pi \times \mathcal{S} \times \mathcal{A} \times \mathcal{S} \times \mathcal{A} :\to [0, 1]$ as follows:

$$P_\nu(\pi|s_t, a_t) = \begin{cases} q_{\pi_\theta,\nu}(s, a) & \text{if } \pi = \pi_E \\ 1 - P_\nu(\pi_E|s_t, a_t) & \text{otherwise} \end{cases} \tag{12}$$

$$P_{\omega\nu\gamma}(\pi|s_t, a_t, s_{t+1}, a_{t+1}) = \begin{cases} r_\omega(s_t, a_t) q_{\pi_\theta,\nu}^\gamma(s_{t+1}, a_{t+1}) & \text{if } \pi = \pi_E \\ 1 - P_{\omega\nu\gamma}(\pi_E|s_t, a_t, s_{t+1}, a_{t+1}) & \text{otherwise} \end{cases} \tag{13}$$

Using $P_\nu$ and $P_{\omega\nu\gamma}$, the loss to satisfy Equation (11) can be rewritten as follows:

$$\begin{aligned} L(\omega, \nu, \theta) &= \mathbb{E}_{s_{t+1}\sim\mathcal{T}, a_{t+1}\sim\pi_\theta}\left[\log P_{\omega\nu\gamma}(\pi_E|s_t, a_t, s_{t+1}, a_{t+1})\right] - \log P_\nu(\pi_E|s_t, a_t) \\ &\leq \log \frac{\mathbb{E}_{s_{t+1}\sim\mathcal{T}, a_{t+1}\sim\pi_\theta}\left[P_{\omega\nu\gamma}(\pi_E|s_t, a_t, s_{t+1}, a_{t+1})\right]}{P_\nu(\pi_E|s_t, a_t)} \end{aligned} \tag{14}$$

We use Jensen's inequality with the concave property of logarithm in Equation (14). Now we see that the loss $L(\omega, \nu, \theta)$ is bounded by the log likelihood ratio between the two Bernoulli distributions $P_\nu$ and $P_{\omega\nu\gamma}$, and $L(\omega, \nu, \theta) = 0$ if $P_\nu(\pi_E|s_t, a_t) = \mathbb{E}_{s_{t+1}\sim\mathcal{T}, a_{t+1}\sim\pi_\theta}\left[P_{\omega\nu\gamma}(\pi_E|s_t, a_t, s_{t+1}, a_{t+1})\right]$. In the end, learning the approximator $Q_{\pi_\theta,\nu}$ turns out to be matching the two Bernoulli distributions. A natural way to measure the difference between two probability distributions is divergence. We choose Jensen-Shannon (JS) divergence to measure the difference because we empirically found it works better, and thereby the objective to optimize $Q_{\pi_\theta,\nu}$ can be written as follows:

$$\arg\min_\nu \mathbb{E}\left[D_{JS}\left(P_\nu(\cdot|s_t, a_t) \,\|\, \mathbb{E}_{s_{t+1}\sim\mathcal{T}, a_{t+1}\sim\pi_\theta}\left[P_{\omega\nu\gamma}(\cdot|s_t, a_t, s_{t+1}, a_{t+1})\right]\right)\right] \tag{15}$$

where $D_{JS}$ denotes JS divergence between two Bernoulli distributions.

### 3.1.3 VALUE ESTIMATION WITHOUT REWARD LEARNING

Suppose the optimal reward function $R_{\omega^*}(s, a) = \log r_{\omega^*}(s, a)$ for the objective (8) can be obtained, the Bellman equation (11) can be rewritten as follows:

$$\log r_{\omega^*}(s_t, a_t) = \log q_{\pi_\theta,\nu}(s_t, a_t) - \mathbb{E}_{s_{t+1}\sim\mathcal{T}, a_{t+1}\sim\pi_\theta}\left[\log q_{\pi_\theta,\nu}^\gamma(s_{t+1}, a_{t+1})\right] \tag{16}$$

Recall that IRL objective (8) aims at assigning $r_{\omega^*}(s_t, a_t) = 1$ for $(s_t, a_t) \in \mathcal{S}_E \times \mathcal{A}_E$ and $r_{\omega^*}(s_t, a_t) = 0.5$ for $(s_t, a_t) \in \mathcal{S}_\pi \times \mathcal{A}_\pi$ where $\pi \in \Pi \setminus \{\pi_E\}$. Therefore, the objective (8) is rewritten as the following objective using the Bellman equation (11) :

$$\begin{aligned} \arg\min_\nu \; &\mathbb{E}_{s_t\sim\rho_{\pi_E}, a_t\sim\pi_E}\left[\log q_{\pi_\theta,\nu}(s_t, a_t) - \mathbb{E}_{s_{t+1}\sim\mathcal{T}, a_{t+1}\sim\pi_\theta}\left[\log q_{\pi_\theta,\nu}^\gamma(s_{t+1}, a_{t+1})\right]\right] \\ &+ \mathbb{E}_{s_t\sim\rho_\pi, a_t\sim\pi}\left[\log q_{\pi_\theta,\nu}(s_t, a_t) - \mathbb{E}_{s_{t+1}\sim\mathcal{T}, a_{t+1}\sim\pi_\theta}\left[\log \{q_{\pi_\theta,\nu}^\gamma(s_{t+1}, a_{t+1})/2\}\right]\right] \end{aligned} \tag{17}$$

Thus, $r_{\omega^*}$ can be obtained by the Bellman equation (16) as long as the solution for the objective (17) can be obtained. We optimize $q_{\pi_\theta,\nu}(s_t, a_t)$ in the same way of objective (15) as follows:

$$\begin{aligned} \arg\min_\nu \; &\mathbb{E}_{s_t\sim\rho_{\pi_E}, a_t\sim\pi_E}\left[D_{JS}\left(P_\nu(\cdot|s_t, a_t) \,\|\, \mathbb{E}_{s_{t+1}\sim\mathcal{T}, a_{t+1}\sim\pi_\theta}\left[P_\nu^\gamma(\cdot|s_{t+1}, a_{t+1})\right]\right)\right] \\ &+ \mathbb{E}_{s_t\sim\rho_\pi, a_t\sim\pi}\left[D_{JS}\left(P_\nu(\cdot|s_t, a_t) \,\|\, \mathbb{E}_{s_{t+1}\sim\mathcal{T}, a_{t+1}\sim\pi_\theta}\left[P_\nu^\gamma(\cdot|s_{t+1}, a_{t+1})/2\right]\right)\right] \end{aligned} \tag{18}$$

We use $P_\nu^\gamma$ instead of $P_{\omega\nu\gamma}$ in objective (18) unlike the objective (15). Thus, we omit reward learning that the existing AIL methods require, while learning $q_{\pi_\theta,\nu}(s_t, a_t)$ to obtain $r_{\omega^*}$.

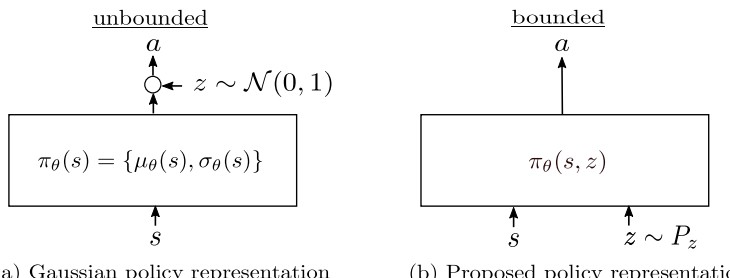

(a) Gaussian policy representation      (b) Proposed policy representation

Figure 1: Functional representations of (a) Gaussian policy and (b) proposed policy.

---

**Algorithm 1** Overview of our IL algorithm

---
1: Initialize parameters $\nu$ and $\theta$.
2: Fulfill a buffer $\mathcal{B}_{\pi_E}$ by the expert demonstrations and initialize the replay buffer $\mathcal{B}_\beta \leftarrow \emptyset$.
3: **for** episode = 1, M **do**
4:      Initialize time-step $t = 0$ and receive initial state $s_0$
5:      **while not** terminate condition **do**
6:          Execute an action $a_t = \pi_\theta(s_t, z)$ with $z \sim P_z$ and observe new state $s_{t+1}$
7:          Store a state-action triplet $(s_t, a_t, s_{t+1})$ in $\mathcal{B}_\beta$.
8:          $t = t + 1$
9:      **end while**
10:      **for** $u = 1, t$ **do**
11:          Sample mini-batches of triplets $(s_{t'}^i, a_{t'}^i, s_{t'+1}^i)$ and $(s_{t'}^j, a_{t'}^j, s_{t'+1}^j)$ from $\mathcal{B}_{\pi_E}$ and $\mathcal{B}_\beta$, respectively.
12:          Update $\nu$ using the sampled gradients of (18) w.r.t $\nu$ using $(s_{t'}^i, a_{t'}^i, s_{t'+1}^i)$ and $(s_{t'}^j, a_{t'}^j, s_{t'+1}^j)$.
13:          Sample a mini-batch of triplets $(s_{t'}^k, a_{t'}^k, s_{t'+1}^k)$ from $\mathcal{B}_\beta$.
14:          Update $\theta$ using the sampled policy gradients (6) using $(s_{t'}^k, a_{t'}^k, s_{t'+1}^k)$.
15:      **end for**
16: **end for**

---

### 3.2 STOCHASTIC POLICY FUNCTION WITH BOUNDED OUTPUTS

Recall that the aim of IL is to imitate the expert behavior. It can be summarized that IL attempts to obtain a generative model the expert has over $\mathcal{A}$ conditioned on states in $\mathcal{S}$. We see that the aim itself is equivalent to that of conditional generative adversarial networks (cGANs) (Mirza & Osindero, 2014). The generator of cGANs can generate stochastic outputs of which range are bounded. As mentioned in Section 1, bounding action values is expected to make the problem easier to solve and make the training faster. In the end, we adopt the form of the conditional generator to represent the stochastic learner policy $\pi_\theta(s, z)$. The typical Gaussian policy and the proposed policy representations with neural networks are described in Figure 1.

### 3.3 OFF-POLICY ACTOR-CRITIC IMITATION LEARNING

Algorithm.1 shows the overview of our off-policy actor-critic imitation learning algorithm.

To learn the value function approximator $Q_{\pi_\theta, \nu}$, we adopt a behavior policy $\beta$ as $\pi$ in the second term in objective (18) We employ a mixture of the past learner policies as $\beta$ and a replay buffer $\mathcal{B}_\beta$ (Mnih et al., 2015) to perform sampling $s_t \sim \rho_\pi$, $a_t \sim \pi$ and $s_{t+1} \sim \mathcal{T}$. The buffer $\mathcal{B}_\beta$ is a finite cache and stores the $(s_t, a_t, s_{t+1})$ triplets in a first-in-first-out manner while the learner interacts with the environment.

The approximator $Q_{\pi_\theta, \nu}(s_t, a_t) = \log q_{\pi_\theta, \nu}(s_t, a_t)$ takes $(-\infty, 0]$. With the approximator, using the gradient (5) to update the learner policy always punish (or ignore) the learner's actions. Instead, we adopt the gradient (6) which directly uses Jacobian of $Q_{\pi_\theta, \nu}$.

As do off-policy RL methods such as Mnih et al. (2015) and Lillicrap et al. (2015), we use the target value function approximator, of which parameters are updated to track $\nu$ , to optimize $Q_{\pi_\theta, \nu}$. We update $Q_{\pi_\theta, \nu}$ and $\pi_\theta$ at the end of each episode rather than following each step of interaction.

## 4  RELATED WORK

In recent years, the connection between generative adversarial networks (GAN) (Goodfellow et al., 2014) and IL has been pointed out (Ho & Ermon, 2016; Finn et al., 2016a). Ho & Ermon (2016) show that IRL is a dual problem of RL which can be deemed as a problem to match the learner's occupancy measure (Syed et al., 2008) to that of the expert, and that a choice of regularizer for the cost function yields an objective which is analogous to that of GAN. Their algorithm, namely GAIL, has become a popular choice for IL and some extensions of GAIL have been proposed (Baram et al., 2017; Hausman et al., 2017; Li et al., 2017). However, those extensions have never addressed reducing the number of interactions during training.

There has been a few attempts that try to improve the sample complexity in IL literatures, such as Guided Cost Learning (GCL) (Finn et al., 2016b). However, those methods have worse imitation capability in comparison with GAIL, as reported by Fu & Levine (2017). As detailed in section 5, our algorithm have comparable imitation capability to GAIL while improving the sample complexity.

Hester & Osband (2017) proposed an off-policy algorithm using the expert demonstration. They address problems where both demonstration and hand-crafted rewards are given. Whereas, we address problems where only the expert demonstration is given.

There is another line of IL works where the learner can ask the expert which actions should be taken during training, such as DAgger (Ross & Bagnell, 2011), SEARN (Daumé & Marcu, 2009), SMILe (Ross & Bagnell, 2010), and AggreVaTe (Ross & Bagnell, 2014). As opposed to those methods, we do not suppose that the learner can query the expert during training.

## 5  EXPERIMENTS

In our experiments, we aim to answer the following three questions:

Q1. Can our algorithm enable the learner to imitate the expert behavior?

Q2. Is our algorithm more sample efficient than BC in terms of the expert demonstration?

Q3. Is our algorithm more efficient than GAIL in terms of the training time?

### 5.1  SETUP

To answer the questions above, we use five physics-based control tasks that are simulated with MuJoCo physics simulator (Todorov et al., 2012). See Appendix A for the description of each task. In the experiments, we compare the performance of our algorithm, BC, GAIL, and GAIL initialized by BC[2][3]. The implementation details can be found in Appendix B. We train an agent on each task by TRPO (Schulman et al., 2015a) using the rewards defined in the OpenAI Gym (Brockman et al., 2016), then we use the resulting agent with a stochastic policy as the expert for the IL algorithms. We store $(s_t, a_t, s_{t+1})$ triplets during the expert demonstration, then the triplets are used as training samples in the IL algorithms. In order to study the sample efficiency of the IL algorithms, we arrange two setups. The first is *sparse sampling setup*, where we randomly sample 100 $(s_t, a_t, s_{t+1})$ triplets from each trajectory which contains 1000 triplets. Then we perform the IL algorithms using datasets that consist of several 100s triplets. Another setup is *dense sampling setup*, where we use full $(s_t, a_t, s_{t+1})$ triplets in each trajectory, then train the learner using datasets that consist of several trajectories. If an IL algorithm succeeds to imitate the expert behavior in the dense sampling setup whereas it fails in the sparse sampling setup, we evaluate the algorithm as sample inefficient in terms of the expert demonstration. The performance of the experts and the learners are measured by cumulative reward they earned in a trajectory. We run three experiments on each task, and measure the performance during training.

---

[2]Ho & Ermon (2016) suggest that initializing policy parameters with BC could significantly improve learning speed, but they did not show any results of such initialization.

[3]We also conducted the same comparison with MGAIL(Baram et al., 2017) using an online available code provided by the authors. Unfortunately, we never reproduced the same performance as reported in the paper on all tasks except for Hopper-v1 even if we followed the author's advice.

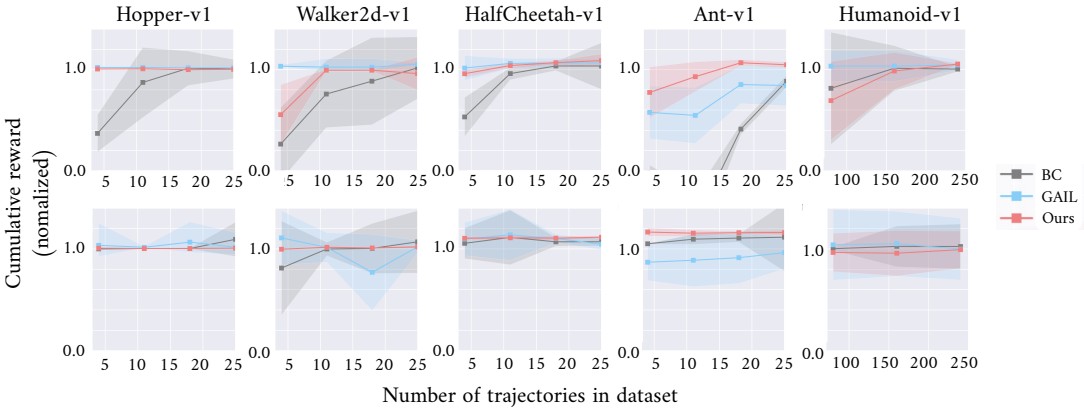

Figure 2: The cumulative reward (normalized) vs. the number of trajectories in a dataset. The results in sparse and dense sampling setup are depicted on top and bottom row, respectively.

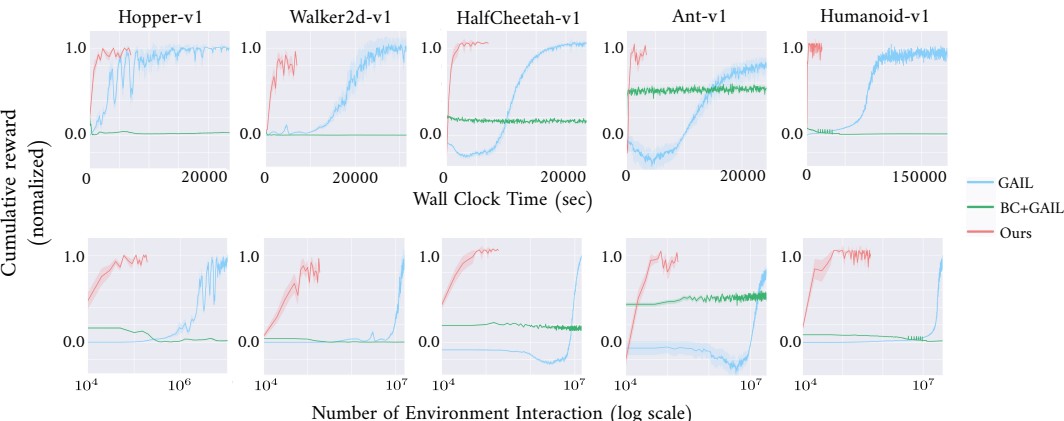

Figure 3: The cumulative reward (normalized) vs. training time (top row) and the number of environment interactions (bottom row).

## 5.2 RESULTS

Figure 2 shows the experimental results in both sparse and dense sampling setup. In comparison with GAIL, our algorithm marks worse performance on Walker2d-v1 and Humanoid-v1 with the datasets of the smallest size in sparse sampling setup, better performance on Ant-v1 in both setups, and competitive performance on the other tasks in both setups. Overall, we conclude that our algorithm is competitive with GAIL with regards to performance. That is, our algorithm enables the learner to imitate the expert behavior as well as GAIL does. BC imitates the expert behavior successfully on all tasks in the dense sampling setup. However, BC often fails to imitate the expert behavior in the sparse sampling setup with smaller datasets. Our algorithm achieves better performance than BC does all over the tasks. It shows that our algorithm is more sample efficient than BC in terms of the expert demonstration.

Figure 3 shows the performance plot curves over validation rollouts during training in the sparse sampling setup. The curves on the top row in Figure 3 show that our algorithm denoted by Ours trains the learner more efficiently than GAIL does in terms of training time. In addition, the curves on the bottom row in Figure 3 show that our algorithm trains the learner much more efficiently than GAIL does in terms of the environment interaction. As opposed to Ho & Ermon (2016) suggestion, GAIL initialized by BC (BC+GAIL) does not improve the sample efficiency, but rather harms the leaner's performance significantly.

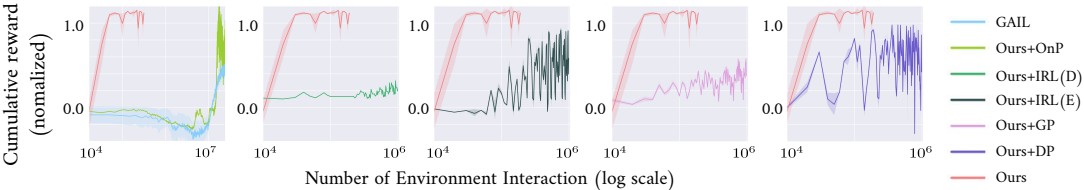

Figure 4: The cumulative reward (normalized) vs. the number of environment interactions on Ant-v1 in the ablation experiment.

### 5.3 ABLATION EXPERIMENTS

We conducted additional ablation experiments to demonstrate that our proposed method described in Section.3 improves the sample efficiency. Figure 4 shows the ablation experimental results on Ant-v1 task. Ours+OnP, which denotes an on-policy variant of Ours, requires 100 times more interactions than Ours. The result with Ours+OnP suggests that adopting off-policy learning scheme instead of on-policy one significantly improves the sample efficiency. Ours+IRL(D) and Ours+IRL(E) are variants of Ours that learn value function approximators using the learned reward function with the objective (2) and (8), respectively. The result with Ours+IRL(D) and Ours+IRL(E) suggests that omitting the reward learning described in 3.1 makes the training stable and faster. The result with Ours+GP, which denotes a variant of Ours that adopts the Gaussian policy, suggests that bounding action values described in 3.2 makes the training faster and stable. The result with Ours+DP, which denotes a variant of Ours that has a deterministic policy with fixed input noises, fails to imitate the expert behavior. It shows that the input noise variable $z$ in our algorithm plays a role to obtain stochastic policies.

## 6 CONCLUSION

In this paper, we proposed a model-free IL algorithm for continuous control. Experimental results showed that our algorithm achieves competitive performance with GAIL while significantly reducing the environment interactions.

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

## A    DETAILED DESCRIPTION OF EXPERIMENT

Table 1 summarizes the description of each task, the performance of an agent with random policy, and the performance of the experts.

Table 1: Description of each task, an agent's performance with random policy, and the performance of the experts. $\mathbf{dim}(\mathcal{S})$ and $\mathbf{dim}(\mathcal{A})$ denote dimensionality of state and action spaces respectively.

| Task | $\mathbf{dim}(\mathcal{S})$ | $\mathbf{dim}(\mathcal{A})$ | Random Policy | Expert's Performance |
|---|---|---|---|---|
| HalfCheetah-v1 | 17 | 6 | -282.43 $\pm$ 79.53 | 4130.22 $\pm$ 75.51 |
| Hopper-v1 | 11 | 3 | 14.47 $\pm$ 7.96 | 3778.05 $\pm$ 3.34 |
| Walker2d-v1 | 17 | 6 | 0.57 $\pm$ 4.59 | 5510.67 $\pm$ 74.44 |
| Ant-v1 | 111 | 8 | -69.68 $\pm$ 111.10 | 4812.93 $\pm$ 122.26 |
| Humanoid-v1 | 376 | 17 | 122.87 $\pm$ 35.11 | 10395.51 $\pm$ 205.81 |

## B    IMPLEMENTATION DETAILS

We implement our algorithm using two neural networks with two hidden layers. Each network represents $\pi_\theta$ and $q_\nu$. For convenience, we call those networks for $\pi_\theta$ and $q_\omega$ as policy network (PN) and Q-network (QN), respectively. PN has 100 hidden units in each hidden layer, and its final output is followed by hyperbolic tangent nonlinearity to bound its action range. QN has 500 hidden units in each hidden layer and a single output is followed by sigmoid nonlinearity to bound the output between [0,1]. All hidden layers are followed by leaky rectified nonlinearity (Maas et al., 2013). The parameters in all layers are initialized by Xavier initialization (Glorot & Bengio, 2010). The input of PN is given by concatenated vector representations for the state $s$ and noise $z$. The noise vector, of which dimensionality corresponds to that of the state vector, generated by zero-mean normal distribution so that $z \sim P_z = \mathcal{N}(0,1)$. The input of QN is given by concatenated vector representations for the state $s$ and action $a$. We employ RMSProp (Hinton et al., 2012) for learning parameters with a decay rate 0.995 and epsilon $10^{-8}$. The learning rates are initially set to $10^{-3}$ for QN and $10^{-4}$ for PN, respectively. The target QN with parameters $\nu'$ are updated so that $\nu' = 10^{-3}*\nu+(1-10^{-3})*\nu'$ at each update of $\nu$. We linearly decrease the learning rates as the training proceeds. We set mini-batch size of $(s_t, a_t, s_{t+1})$ triplets 64, the replay buffer size $|\mathcal{B}_\beta| = 15000$, and the discount factor $\gamma = 0.85$. We sample 128 noise vectors for calculating empirical expectation $\mathbb{E}_{z \sim P_z}$ of the gradient (6). We use publicly available code (`https://github.com/openai/imitation`) for the implementation of GAIL and BC. Note that, the number of hidden units in PN is the same as that of networks for GAIL. All experiments are run on a PC with a 3.30 GHz Intel Core i7-5820k Processor, a GeForce GTX Titan GPU, and 32GB of RAM.

