# OpenReview forum: "Sample Efficient Imitation Learning for Continuous Control"
_ICLR.cc/2019/Conference_

### Official Review · AnonReviewer1 · 2018-10-28
**Efficient imitation learning**

**Rating:** 5
**Confidence:** 5

**Review:**

This paper proposes a new method to imitate expert efficiently. The paper first proposes a way to compute reward function from expert demonstration and uses the log probability to represent this reward function.  Then they find a form of bellman equation that can optimize the reward stably. After the 'Q learning without IRL', an off-policy RL off-pac is applied. So this paper achieves comparable results to GAIL but uses much less data amount.

clarity:
This paper is clearly written.

originality:
This paper is original.

pros:
Comparable performance with GAIL.
Better performance than Behavioral Cloning
New way of using demonstrations

cons:
Although both the method and the experiments look promising, there is a very simple yet competitive baseline missing. This baseline is also mentioned in the original GAIL paper: you initialize GAIL with BC, and then train GAIL. That's the baseline for a set of fair comparison.

---

### Official Review · AnonReviewer3 · 2018-11-04
**Hard to read, probably overfits ?**

**Rating:** 5
**Confidence:** 5

**Review:**

The paper proposes a method for imitation learning via inverse reinforcement learning based on a specific modeling of the reward. It is modeled as the log probability of a state action pair to belong to the expert policy. It models this distribution as a Bernoulli one and thus it reduces the IRL problem to a classification task. The global method also uses an off-policy algorithm to learn the value function of the current agent policy to improve sample efficiency. The method is tested on a set of continuous control tasks such as walker, hopper or humanoid.

I think the paper has several flaws. First, I found the paper not very well written and organized. It is hard to read. It uses some terminology in a way that is different from the rest of the littérature (such as Q-learning as learning the Q-function of the expert policy instead of using the  optimal Bellman operator (even if the expert is supposed to be optimal)). I also think that the related work section is missing a lot of important refs because it really focuses on recent papers while imitation learning has a long history.

Yet, my main concern is that the proposed method seems to reduce to a classification problem to me and is likely to suffer from the same issues than the supervised learning method (AKA behavior cloning). It probably overfits a lot and there is nothing in the experiments that shows how robust is the method to perturbations. In a discrete world, this method would ideally place a reward of 1 in every state visited by the expert and 0 elsewhere which is very likely to overfit and result in unstable behaviors in the presence of noise etc. I would like to see experiments showing robustness.

The experiments are also a bit strange since the learning is stopped early for the proposed method. Is it because the learning is unstable ?

---

### Official Review · AnonReviewer4 · 2018-11-07
**Nice idea, nice results, but paper needs work.**

**Rating:** 7
**Confidence:** 5

**Review:**

This paper proposed an imitation learning algorithm that achieves competitive results with GAIL, while requiring significantly fewer interactions with the environment.

I like the method proposed in this paper. It seems similar to ideas in this concurrent submission: https://openreview.net/forum?id=B1excoAqKQ

However, the paper is a bit difficult to read. The proposed method is made up of several changes compared to the baselines (e.g. using Q-learning without IRL instead of IRL, using off-policy learning, using conditioning to obtain a stochastic policy) but motivation for each component is presented late within the paper. The terminology used to describe these components is a bit confusing. Also some math is presented without intuitive descriptions.

I’d like to see more ablations performed: there are three main changes compared to GAIL, but an ablation is only performed for the stochastic policy. It would be interesting to tease out what is more important, off-policy learning, or bypassing IRL.

---

### Official Review · AnonReviewer2 · 2018-11-07
**Good results, but needs ablations to clearly identify contributions and improved presentation**

**Rating:** 5
**Confidence:** 4

**Review:**

Summary/contributions:

The primary aim of this paper is to improve the sample efficiency of GAIL (Ho et al. 2016). The claimed contributions can be summarized by consisting of 1) replacing TRPO (which was used in the original paper) with a off-policy RL with a modified reward, 2) using a policy parameterizing where the noise is used as an input rather than at the output. While conceptually simple, this paper contributes a method that shows improved sample efficiency on a series of benchmark mujoco tasks, which has practical implications for real world environments.

Pros:
- a simple idea with good empirical results that would be of interest to the community

Cons:
- (extremely) unclear presentation which hinders the message of the paper.
- the novelty of the approach is somewhat limited

Justification for score:
I gave my rating based upon the following considerations. The approach in this paper makes sense from a practical perspective and presents strong results. However, the experiments in the paper do not clearly identify which components of their method lead to their improved performance (i.e., an ablation on their stated contributions). The writing is also extremely poor. The paper makes use of non-standard notation (in relation to the prior work which it builds on) and unusual terminology. Overall however, I am on the fence about this paper, since I recognize the good results presented in this paper, in addition to the timely nature of the idea (there are at least two concurrent submissions that I am aware of that are similar).

Other:
- I would appreciate if the related work discussed prior off-policy methods that use demonstrations (e.g Hester et al. 2017)
- The paper has a large number of ungrammatical sentences and unidiomatic expressions.

---

### Meta-Review · Area_Chair1 · 2018-12-14
**A simple but potentially impactful idea and a manuscript that has greatly improved since submission.**

**Confidence:** 4
**Recommendation:** Accept (Poster)

**Metareview:**

The paper proposes a simple method for improving the sample efficiency of GAIL, essentially a way of turning inverse reinforcement learning into classification. As reviewers noted, the method is based on a simple idea with potentially broad applicability.

Concerns were raised about the multiple components of the system and what each contributed, and missing pointers to the literature. A baseline wherein GAIL is initialized with behaviour cloning, although only suggested but not tried in previous works. The authors did, however, attempt this setting and found it to hurt, not help, performance. I find this surprising and would urge the authors to validate that this isn't merely an uninteresting artifact of the setup, however I commend the authors for trying it and don't believe that a surprising result in this regard is a barrier to publication.

As several reviewers did not provide feedback on revisions addressing their concerns, this Area Chair was left to determine to a large degree whether or not reviewer concerns were in fact addressed.  I thank AnonReviewer4 for revisiting their review towards the end of the period, and concur with them that many of the concerns raised by reviewers have indeed been adequately dealt with.